# Effects of passive heat stress and recovery on human cognitive function: An ERP study

**Hiroki Nakata** [1], **Ryusuke Kakigi**[2], **Manabu Shibasaki**[1] *

**1** Department of Health Sciences Health Sciences, Faculty of Human Life and Environment, Nara Women's University, Nara City, Japan, **2** Department of Integrative Physiology, National Institute for Physiological Sciences, Okazaki, Japan

* shiba@cc.nara-wu.ac.jp

**Data Availability Statement:** All relevant data are within the paper and its Supporting Information files.

**Funding:** This study was supported by a Japan Society for the Promotion of Science KAKENHI

## Abstract

Using event-related potentials (ERPs), we investigated the effects of passive heat stress and recovery on the human cognitive function with Flanker tasks, involving congruent and incongruent stimuli. We hypothesized that modulation of the peak amplitude and latency of the P300 component in ERP waveforms would differ with task difficulty during passive heat stress and recovery. Subjects performed the Flanker tasks before (Pre), at the end of whole body heating (Heat: internal temperature increase of ~1.2°C from the pre-heat baseline), and after the internal temperature had returned to the pre-heat baseline (Recovery). The internal temperature was regulated by a tube-lined suit by perfusing 50°C water for heat stress and 25°C water for recovery immediately after the heat stress. Regardless of task difficulty, the reaction time (RT) was shortened during Heat rather than Pre and Recovery, and standard deviations of RT (i.e., response variability) were significantly smaller during Heat than Pre. However, the peak amplitudes of the P300 component in ERPs, which involved selective attention, expectancy, and memory updating, were significantly smaller during Heat than during Pre, suggesting the impairment of neural activity in cognitive function. Notably, the peak amplitudes of the P300 component were higher during Recovery than during Heat, indicating that the impaired neural activity had recovered after sufficient whole-body cooling. An indicator of the stimulus classification/evaluation time (peak latency of P300) and the RT were shortened during Heat stress, but such shortening was not noted after whole-body cooling. These results suggest that hyperthermia affects the human cognitive function, reflected by the peak amplitude and latency of the P300 component in ERPs during the Flanker tasks, but sufficient treatment such as whole-body cooling performed in this study can recover those functions.

## Introduction

Regarding some recent review articles on hyperthermia, central fatigue induced by hyperthermia impairs the voluntary force production and neuromuscular function, resulting from impairments in psychological and physiological factors in the central nervous system [see reviews, 1–3]. It is possible that symptoms of heat exhaustion involving faintness, dizziness,

Grant-in-Aid for Scientific Research B 18H03166 (to MS), and Scientific Research C 19K11576 (to HN).

**Competing interests:** No authors have competing interests.

and headache are subjectively related to declines in the cognitive function. However, the detailed mechanisms for the declines with hyperthermia remain unclear because of methodological discrepancies including research approaches such as vigilance and monitoring performance tests, tracking or cognitive/skilled tasks, as well as combined stresses due to environmental and physiological conditions [2]. An objective and reproducible index is needed.

Several non-invasive recording methods have been used to investigate the human cognitive function, such as functional magnetic resonance imaging (fMRI), functional near-infrared spectroscopy (fNIRS), and transcranial magnetic stimulation (TMS). Event-related potentials (ERPs), obtained by time-locked averaging electroencephalography (EEG) with high temporal resolution, have been used to evaluate the higher cognitive function that involves selective attention, expectancy, and memory updating [4]. The amplitude of each component of ERPs reflects the intensity of neurocognitive processing [5]. Larger amplitudes indicate more neural excitability and/or neuronal circuits related to endogenous and exogenous components in the amount of attentional resources [4, 6]. On the other hand, smaller or decreased amplitudes have been observed in untrained, aged, or exhausted individuals [7, 8].

Recently, we examined the changes in the peak amplitudes of P300 components of ERPs, which were measured at 300–600 ms after stimulus onset, to assess cognitive function during passive heat stress and in a hot environment [9–11]. We found the reduced peak amplitude of P300 during heat stress, suggesting impairment of the cognitive function. In those studies, two different modalities (auditory and somatosensory stimuli) and paradigms (oddball and Go/No-go tasks) were adopted, and they showed similar reductions in the amplitudes of P300 components while subjects were being subjected to hyperthermic conditions. Interestingly, we also reported that the peak amplitude of P300 remained reduced during subsequent whole-body cooling for 3 min [9]. Moreover, in their follow-up study, while maintaining hyperthermia, subjects' face/head were cooled with a fan and ice pack to induce a thermally comfortable condition [10]. However, the peak amplitude of P300 remained reduced (i.e., did not recover).

Taking these findings into consideration, two issues should be resolved to clarify the detailed mechanisms explaining the effects of hyperthermia on the cognitive function. The first involves task difficulty and the sensory modality. Shibasaki and colleagues used a simple cognitive task with "auditory" oddball and "somatosensory" Go/No-go paradigms, whereby subjects pushed a button in response to a target stimulus when presented with a continuous and random series of target and non-target stimuli. We considered that such effects would be related to the task difficulty or sensory modality. Thus, we chose "visual" Flanker tasks, which were used to examine the neural systems that resolve the conflict among response options [12]. In these tasks, a central target stimulus is presented simultaneously with distractor stimuli (flankers) and subjects are requested to respond according to the target and ignore the flankers. In general, the reaction time (RT) is shorter with Congruent stimuli than Incongruent stimuli [13, 14], indicating different levels of task difficulty [13, 14]. The second involves recovery after heat stress. As mentioned above, a short period of whole-body cooling after heat stress and face/head cooling during heat stress did not lead to full recovery of the peak amplitude of P300 [9, 10]. In addition, after a thorough literature research, we failed to identify any study examining the recovery process after heat stress by P300. Providing scientific evidence to support recovery from hyperthermia might be important to develop a methodology in our daily life and sports activities. Thus, we should examine whether sufficient whole-body cooling recovers the cognitive function reflected by the P300 component in ERPs.

Based on this background, the present study used visual Flanker tasks as complicated cognitive tasks, and investigated the effects of passive heat stress and recovery on ERPs.

## Materials and methods

### Subjects

Sixteen subjects (thirteen males and three females; mean age 21.4 ± 1.0 years) participated in this study. The mean body mass and height of the subjects were 71.7 ± 11.3 kg, and 170.5 ± 6.3 cm, respectively. None of the subjects had a history of neurological or psychiatric disorders. The procedures used complied with the Declaration of Helsinki regarding human experimentation, and the study was approved by the Ethics Committee of Nara Women's University, Nara, Japan. All subjects gave written informed consent to participate in the study.

### Procedure

The room temperature was set at 25˚C, but subjects' skin temperatures were controlled using a two-pieced water-perfused suit. Following instrumentation, subjects wore a tube-lined suit (Med-Eng, Ottawa, Ontario, Canada), which covered the entire body except for the head, face, hands, and feet. Then, they rested quietly in a supine position on a hospital bed for ~30 minutes. Thermoneutral conditions were maintained by perfusing 32˚C water through the tube-lined suit. During this period, EEG electrodes were placed on the scalp and earlobes, while other instruments were attached during the equilibration period. The baseline data of ERPs during the Flanker tasks were recorded (1st (Pre) session). Subjects were then exposed to heat stress by perfusing 50˚C water through the suit, and ERPs were recorded after the external ear canal temperature had increased by ~1.2˚C from the pre-heat baseline (Heat session). After the second cognitive task, cold water (25˚C) was then immediately perfused through the suit to decrease the body temperature. When the external ear canal temperature had retuned to the pre-heat stress baseline and whole-body cooling had been performed for at least 30 min, ERPs were recorded (Recovery session) (Fig 1A).

### Flanker task

A Flanker task consisting of five arrowheads was used (Fig 1B). In a standard Flanker task, the subjects are instructed to press a button with their left or right thumbs as quickly as possible, corresponding to the direction of the centrally presented target arrowhead, which is called a 'compatible' setting. For example, when the central arrowhead was directed to the left, the

(A) Schema of experiment

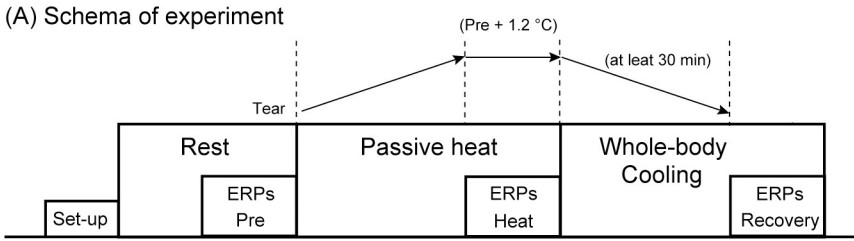

(B) Flanker stimulus

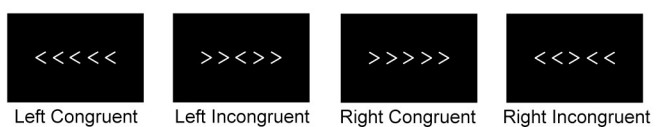

| Left Congruent | Left Incongruent | Right Congruent | Right Incongruent |

**Fig 1.** (A) Stimulus in Flanker task. The subjects were instructed to press a button with their left or right thumbs as quickly as possible, corresponding to the 'opposite' direction of the centrally presented target arrowhead. (B) Schema of the experimental time course. Tear = external canal temperature.

subjects were instructed to press a button with their left thumb. However, after being trained in the compatible setting, subjects were requested to respond in a direction opposite to the target arrowhead, called an 'incompatible' setting. For example, when the central arrowhead was directed to the left, the subjects were instructed to press a button with their right thumb. Arrowheads with white characters were presented for 100 ms on a black background, and the inter-stimulus interval was set at 2900 ms. A run comprised 160 stimuli (i.e., 8 min), which included 40 stimuli for the left congruent arrowhead, 40 stimuli for the right congruent arrowhead, 40 stimuli for the left incongruent arrowhead, and 40 stimuli for the right incongruent arrowhead. The probability of all stimuli was equal, being presented in a random order. Inter-stimulus interval was fixed. Pre, Heat, and Recovery sessions included 160 stimuli, respectively. In a practice run, subjects were instructed to perform the Flanker tasks with 40 stimuli before recording the Pre session. Visual stimuli were presented on a TV monitor (monitor square 17 inch, MITSUBISHI Diamondcrysta RDT174LM) approximately 1 m in front of the subjects using a personal computer programmed by the authors (Hewlett-Packard xw4400 Workstation). The background was black.

## Thermoregulatory and hemodynamic variables

Electrocardiographic and skin temperature probes were attached to monitor the heart rate and mean skin temperature, respectively. The mean skin temperature was indexed from the weighted average of six thermocouples [15]. The external ear canal temperature was measured as an index of the internal temperature using an infrared sensor (Nipro CE Thermo, NIPRO, Japan). The reliability of this device as an index of the internal temperature was confirmed by our previous studies involving simultaneous measurement with changes in the esophageal temperature [10, 16]. It was continuously measured and sampled at 20 Hz via a data acquisition system (MP150, BIOPAC Systems, Santa Barbara, CA, USA) throughout the experiments. Intermittent arterial blood pressure was measured by auscultation of the brachial artery via electrosphygmomanometry (STBP-780, Colin, Tokyo, Japan) before and after each ERP recording. The heart rate was continuously measured and sampled at 20 Hz via a data acquisition system (MP150, BIOPAC Systems, Santa Barbara, CA, USA).

## EEG recordings

ERPs were recorded using Ag/AgCl disk electrodes placed on the scalp at Fz, Cz, Pz, C3 and C4, according to the International 10–20 System. Each scalp electrode was referenced to linked earlobes, which were mathematically calculated and averaged for reference. In order to eliminate eye movements or blinks exceeding 100 μV, an electrooculogram (EOG) was recorded bipolarly with a pair of electrodes placed 2 cm lateral to the lateral canthus of the left eye and 2 cm above the upper edge of the left orbit. Artifacts or noise caused by blinking or sweating were excluded on-line. If the number averaged was less than 35 for each stimulus, additional trials were performed. Impedance was maintained at less than 5 kohm and measured before ERP recording in each session. All electrodes were detached after ERP recordings in Pre and Heat sessions, respectively, to avoid the effect of sweat on the EEG paste. Just before ERP recordings in Heat and Recovery sessions, all electrodes were again attached. All EEG signals were collected on a signal processor (Neuropack MEB-2200 system, Nihon-Kohden, Tokyo, Japan). The bandpass filter of the amplifier was 0.1–50 Hz. The analysis epoch for each ERP stimulus was 800 ms, including a pre-stimulus baseline period of 100 ms. The band-pass filter was set to 0.1–50 Hz and the sampling rate was 1000 Hz. No digital filter was applied off-line. The peak amplitudes and latencies of the P300 component of EEG were measured at 250–600 ms, respectively. If the P300 component showed double peaks, we took the value from the

largest peak. Amplitudes were measured baseline-to-peak. The peak amplitudes and latencies of P300 were assessed using a measuring scale on the Neuropack system with visual inspection.

### Data analyses

A 60-sec average was calculated for the thermoregulatory variables and heart rate before and after each ERP recording. These values and the mean blood pressure (MAP) were averaged between before and after recordings. Thermoregulatory and hemodynamic variables were analyzed by one-way analysis of variance (ANOVA) with repeated measures using the within-subject factor of Session (Pre, Heat, and Recovery). Behavioral data for RT, the standard deviation (SD) of RT (i.e., response variability), and error rates were also separately submitted to a two-way repeated measures ANOVA with Session and Stimulus (Congruent vs. Incongruent) as factors. As for data on the peak amplitudes and latencies of P300, we focused on Cz and Pz of middle electrodes, because the P300 amplitudes were clearly larger at Cz and Pz than at other electrodes. These data were separately submitted to a three-way ANOVA with repeated measures using Session, Stimulus, and Electrode as factors.

For all repeated-measures factors, we tested whether Mauchly's sphericity assumption was violated. If the result of Mauchly's test was significant and the assumption of sphericity was violated, Greenhouse-Geisser adjustment was used to correct sphericity by altering the degrees of freedom using a correction coefficient epsilon. When a significant effect of Session was identified, the Bonferroni post-hoc multiple-comparison was adjusted to identify the specific differences among sessions. When a significant effect of Stimulus was identified, the post-hoc paired-t-test was adjusted to identify the specific difference between Congruent and Incongruent stimuli. We did not adjust the multiple comparison correction for all data. Statistical tests were performed using computer software (SPSS for windows ver. 22.0, SPSS). All statistical analyses were conducted with $\alpha = 0.05$.

## Results

### Thermoregulatory and hemodynamic variables

ANOVAs for mean skin and external canal temperatures showed significant main effects of Session (F (2, 26) = 438.123, $p < 0.001$, $\eta^2 = 0.971$; F (2, 30) = 174.514, $p < 0.001$, $\eta^2 = 0.921$). These variables increased during Heat, and returned to the pre-heating level during Recovery.

ANOVAs for heart rate showed the significant main effect of Session (F (2, 30) = 88.722, $p < 0.001$, $\eta^2 = 0.855$). Heart rate increased during Heat, and almost returned to the pre-heating level during Recovery. ANOVAs for mean blood pressure also showed the significant main effect of Session (F (2, 30) = 6.085, $p < 0.01$, $\eta^2 = 0.289$) (Table 1).

### Behavioral data

ANOVAs for RT showed the significant main effects of Session (F (2, 30) = 24.210, $p < 0.001$, $\eta^2 = 0.617$) and Stimulus (F (1, 15) = 13.863, $p < 0.01$, $\eta^2 = 0.480$). RT differed among sessions, and was shorter with Congruent stimulus than Incongruent stimulus. Post-hoc tests for Session demonstrated that RT was significantly shorter during Heat than Pre, and returned during Recovery with Congruent and Incongruent stimuli (all, $p < 0.05$) (Fig 2A). Post-hoc tests for Stimulus showed that RT was significantly shorter with Congruent stimulus than Incongruent stimulus during Pre, Heat, and Recovery (all, $p < 0.05$).

ANOVAs for SD of RT showed the significant main effects of Session (Greenhouse-Geisser correction: F (1.478, 22.174) = 7.273, $p < 0.01$, $\varepsilon = 0.739$, $\eta^2 = 0.327$) and Stimulus

**Table 1. Thermoregulatory and hemodynamic variables.**

|  | Pre | Heat | Recovery |
|---|---|---|---|
| Tsk (°C) | 33.5 (0.5) | 38.6 (0.3) ***### | 31.7 (1.0) *** |
| Tear (°C) | 37.0 (0.5) | 38.2 (0.3) ***### | 37.0 (0.3) |
| HR (bpm) | 59.8 (10.4) | 92.6 (18.0) ***### | 57.5 (9.3) |
| MAP (mmHg) | 78.9 (8.3) | 81.6 (9.7) | 86.0 (9.0) ** |

Data are expressed as mean (SD). Post-hoc results vs. Pre:

** $p < 0.01$, and

*** $p < 0.001$; Post-hoc results vs. Recovery:

### $p < 0.001$

Tsk = mean skin temperature; Tear = external ear canal temperature; HR = heart rate; MAP = mean arterial blood pressure

($F_{(1, 15)} = 26.871$, $p < 0.001$, $\eta^2 = 0.642$). SD of RT differed among sessions, and was smaller with Congruent stimulus than Incongruent stimulus. Post-hoc tests for Session demonstrated that SD of RT was significantly smaller during Heat than Pre with Congruent and Incongruent stimuli (all, $p < 0.05$) (Fig 2B). Post-hoc tests for Stimulus showed that SD of RT was significantly smaller with Congruent stimulus than Incongruent stimulus during Pre, and Recovery (all, $p < 0.05$).

No significant differences in the error rate were observed among sessions (Fig 2C).

## P300 component

Fig 3 shows grand-averaged ERP waveforms for each session. ANOVAs for the peak amplitude of P300 showed the significant main effects of Session ($F_{(2, 30)} = 6.157$, $p < 0.01$, $\eta^2 = 0.291$) and Stimulus ($F_{(1, 15)} = 4.896$, $p < 0.05$, $\eta^2 = 0.246$). The peak amplitude of P300 differed among sessions, and was larger with Congruent stimulus than Incongruent stimulus. A post-hoc test for Session showed that the peak amplitudes of P300 were significantly smaller during Heat than Pre at Cz and Pz with Congruent and Incongruent stimuli (all, $p < 0.05$) (Fig 4A), and the amplitudes almost returned to the pre-heating level during Recovery with Incongruent stimulus (all, $p < 0.05$) (Fig 4B). Post-hoc tests for Stimulus showed that the peak amplitude of P300 was significantly larger with Congruent stimulus than Incongruent stimulus at Pz during Pre and Heat sessions (all, $p < 0.05$).

ANOVAs for the peak latency of P300 showed the significant main effects of Session ($F_{(2, 30)} = 6.420$, $p < 0.01$, $\eta^2 = 0.300$), and Electrode ($F_{(1, 15)} = 10.938$, $p < 0.01$, $\eta^2 = 0.422$), and Session-Stimulus ($F_{(2, 30)} = 3.785$, $p < 0.05$, $\eta^2 = 0.201$) and Session-Electrode interactions (Greenhouse-Geisser correction: $F_{(1.403, 21.039)} = 4.619$, $p < 0.05$, $\varepsilon = 0.701$, $\eta^2 = 0.235$). A post-hoc test showed that the peak latency of P300 was significantly shorter during Heat and Recovery than during Pre at Cz, and during Heat than Pre at Pz with Congruent stimulus (all, $p < 0.05$) (Fig 4C), and during Heat than Pre and Recovery at Pz with Incongruent stimulus (all, $p < 0.05$) (Fig 4D).

## Discussion

Here, we showed the characteristics of ERPs during passive heat stress and recovery from heat stress with whole-body cooling, when performing visual Flanker tasks. As behavioral data, RT and SD of RT with both Congruent and Incongruent stimuli were shortened and smaller during Heat, and such changes were not noted after whole-body cooling. The peak amplitude of

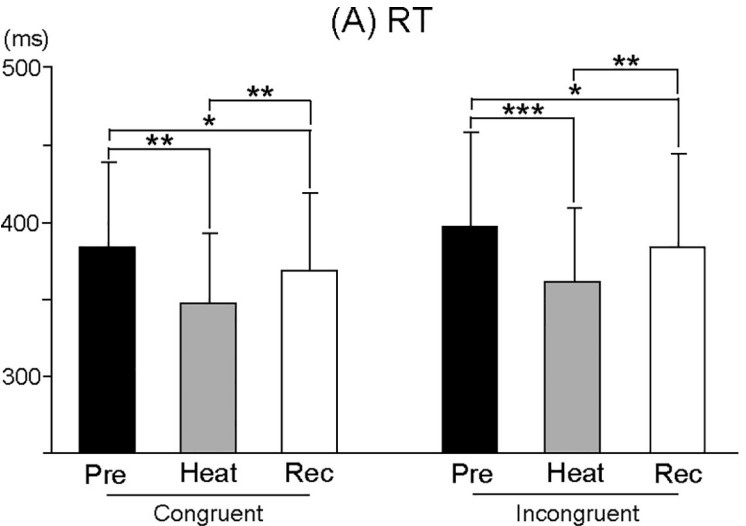

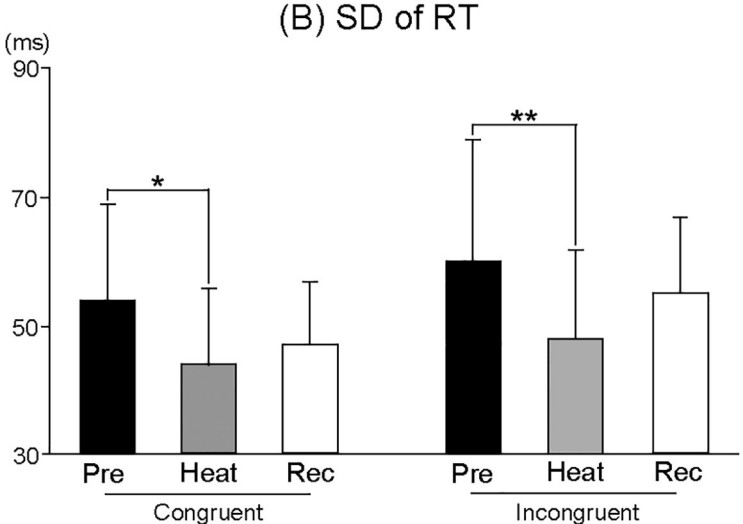

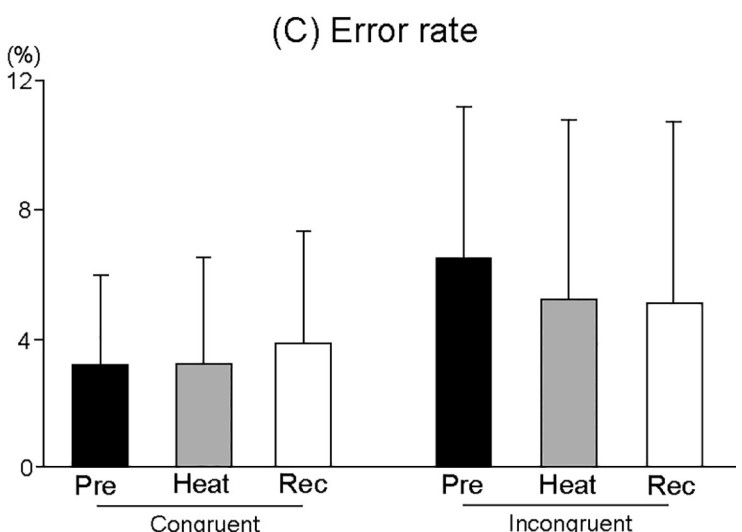

**Fig 2. Behavioral data.** (A) The mean reaction time (RT) for Congruent and Incongruent stimuli of each session. (B) The mean standard deviation (SD) of RT for Congruent and Incongruent stimuli of each session. (C) The mean error rate for Congruent and Incongruent stimuli of each session. The vertical lines indicate SD. * $p < 0.05$; ** $p < 0.01$; *** $p < 0.001$.

P300 decreased, and the peak latency of P300 shortened during Heat. However, the peak amplitude of the P300 recovered, and the shortening in the peak latency of P300 was not found after whole-body cooling.

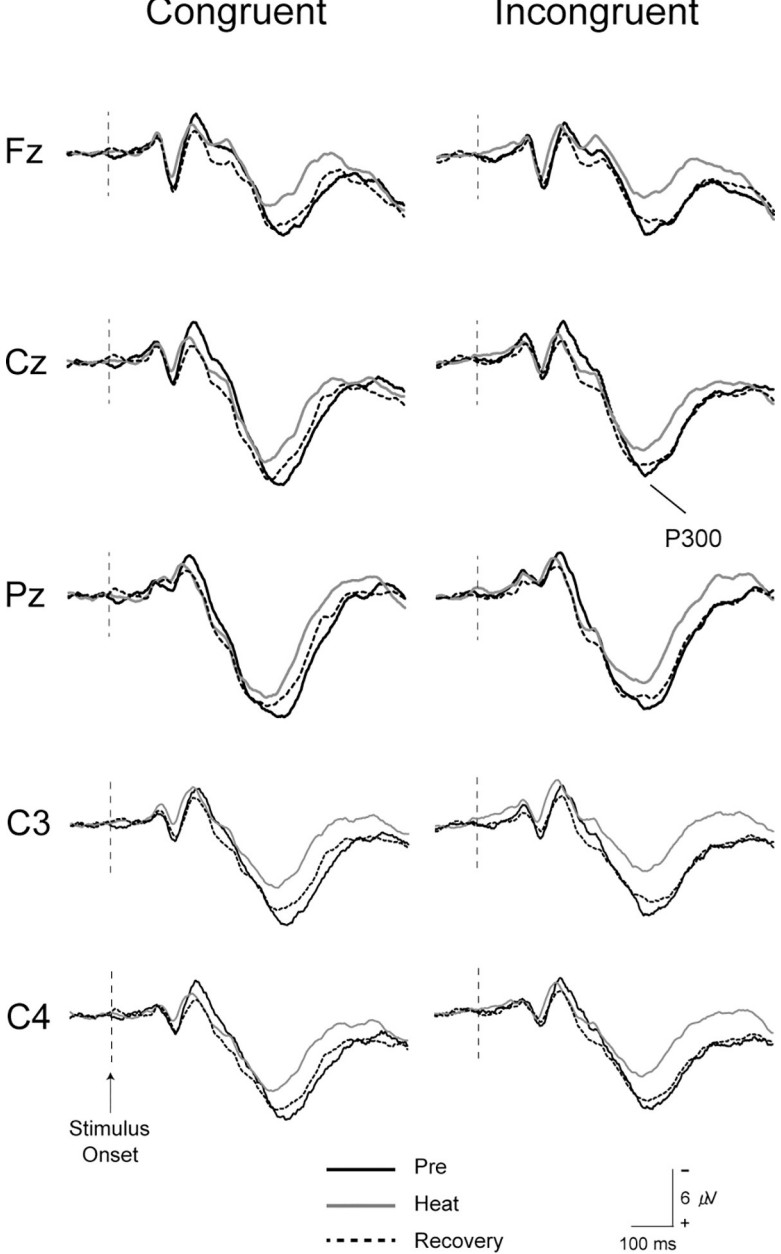

**Fig 3. Grand-averaged EPP waveforms for each session across all subjects.** Black lines indicate waveforms at Pre, gray lines show waveforms at Heat, and dotted lines show waveforms at Recovery.

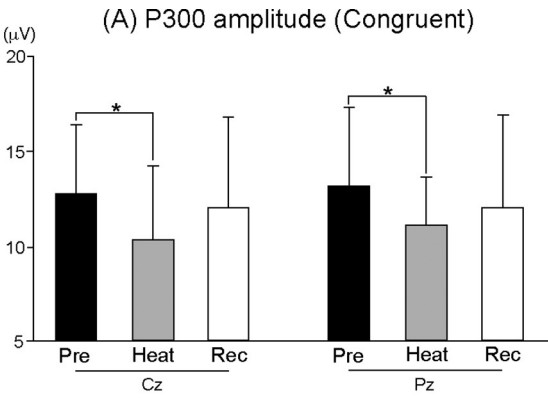

(A) P300 amplitude (Congruent)

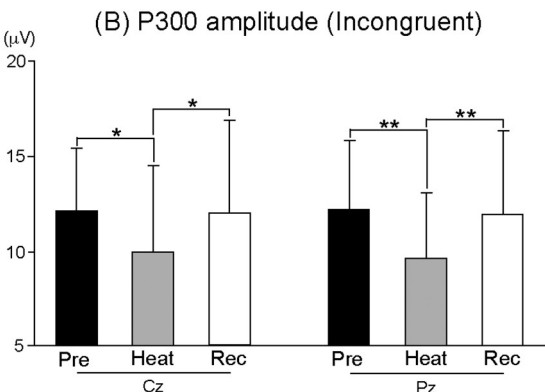

(B) P300 amplitude (Incongruent)

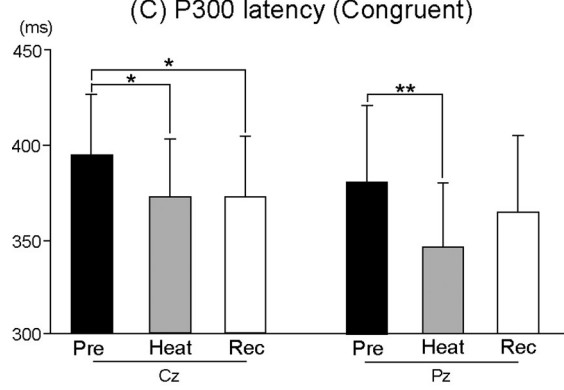

(C) P300 latency (Congruent)

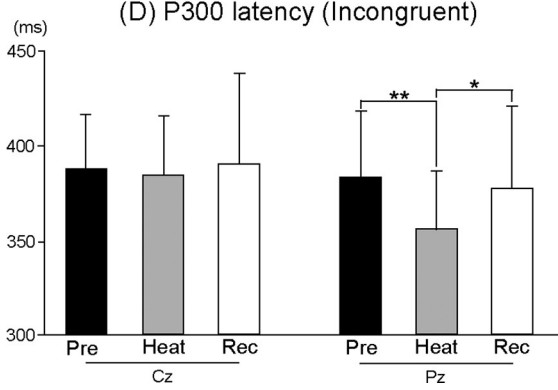

(D) P300 latency (Incongruent)

**Fig 4.** (A) The mean peak amplitude of P300 for Congruent stimulus. (B) The mean peak amplitude of P300 for Incongruent stimulus. (C) The mean peak latency of P300 for Congruent stimulus. (D) The mean peak latency of P300 for Incongruent stimulus. Rec = Recovery session; * p < 0.05; ** p < 0.01.

## The effect of task modality and difficulty on EEG-ERPs

The peak amplitudes of P300 were significantly smaller during Heat than Pre, irrespective of Congruent or Incongruent stimulus (Figs 3 and 4). The amplitude of P300 is proportional to the amount of attentional resources devoted to a given task [17, 18]. For example, the amplitude of P300 was reduced under a dual-task condition whereby the subjects concurrently performed a visuomotor tracking task and somatosensory oddball task, while they performed just the oddball task under an oddball-only condition [19]. Previously, using simple tasks, Shibasaki and colleagues demonstrated that the amplitude of P300 was reduced during passive heat stress in auditory oddball tasks [9], and somatosensory Go/No-go tasks [10]. In the present study, we used 'incompatible' Flanker tasks. Subjects were requested to respond in a direction opposite to the target arrowhead, called an incompatible setting. Therefore, we applied the most complicated task adoptable for EEG-ERPs studies, and showed the reduced amplitude of P300, which was consistent with previous results.

Some previous studies using fMRI reported brain regions during Flanker tasks, and several regions, including the inferior prefrontal cortex, supplementary motor area, anterior cingulate cortex, and posterior parietal cortex, were activated more with Incongruent stimulus than Congruent stimulus [13, 20, 21]. Hazeltine and colleagues suggested that these brain regions were associated with error detection processes [13]. In the present study, the reduced amplitudes of P300 were observed during Heat, irrespective of Congruent or Incongruent stimulus (Fig 4A and 4B). In addition to the task difficulty, the effect of heat stress on the cognitive function using three different tasks (oddball, Go/No-go, and Flanker) and modalities (auditory, somatosensory, and visual) generated the same results. Taking these findings into consideration, the reproducibility of our results suggests that mild hyperthermia decreases the amplitude of P300, and affects the broad cognitive function rather than a specific neural process.

In addition, the reduced amplitude of P300 might be related to electrical noise due to sweating during passive heat stress. However, as mentioned above, to avoid the effect of sweat on the EEG paste, we reattached all electrodes just before ERP recordings in Heat and Recovery sessions. Our previous study using the same procedure as in the present study showed that the amplitudes of some components in somatosensory evoked potentials (SEPs) reduced during passive heat stress, but other components did not [22, 23]. If the presence of sweat influences the SEP waveforms, all SEP components should be distorted. Therefore, we considered that sweat had a reduced effect on the amplitude of P300 in the present study.

## The effectiveness of cooling on impaired cognitive function in hyperthermic individuals

During Recovery from passive heat stress with whole-body cooling, the reduced amplitude of P300 and accelerated latency of P300 returned to the pre-baseline levels (Figs 3 and 4). Interestingly, even accelerated RT and smaller SD of RT during heat stress returned to the pre-heat baseline after whole-body cooling. As mentioned in Introduction, a short period of whole-body cooling after heat stress and face/head cooling during heat stress would not lead to full recovery of the peak amplitude of P300 [9, 10]. These previous studies suggest that brain active potentials are impaired by elevations in internal temperature, even though the skin temperature decreases by these cooling methods. In other words, adequate whole-body cooling is

important for recovery of the higher cognitive function from hyperthermia. This is the first study examining the full recovery process after heat stress by P300 in ERPs. Our results may contribute to the development of a methodology to prevent hyperthermia in daily life and during sports activities. Indeed, face/head cooling led to psychological comfort [10], but our data indicate that recovery of the higher cognitive function reflected by P300, which involves selective attention, expectancy, and memory updating, is caused by adequate whole-body cooling.

We confirmed the reproducibility of the methodology, but from the viewpoint of interpretation, further investigation is needed to clarify the discrepancy of directivity between the amplitude and latency. Behavioral data for RT were significantly shorter during Heat than Pre (Fig 2A), SDs of RT were significantly smaller during Heat than Pre (Fig 2B), and the peak latencies of P300 were also significantly shorter during Heat than Pre (Figs 3 and 4). RT is an important measure for understanding sensorimotor performance in humans [24], and is defined as the time from stimulus onset to a response. Response variability, which is often calculated as SD of RT [25], is used as an index of sustained attention during tasks [26]. The latency of P300 is considered a measure of the stimulus classification speed or stimulus evaluation time [27] and is generally unrelated to response selection processes [28, 29]. The present findings indicate that an increased internal temperature accelerates the classification/evaluation time in Flanker tasks, and drives sustained attention. The interpretation of these results requires special attention. We considered that acceleration of RT and the latency of P300 and smaller SD of RT (response variability) were related to specific effective connectivity in neural networks with an increasing internal temperature, which differs from the theory of structural connectivity [see a review, 30]. Kao et al. [31] recently reported that the peak amplitude of P300 in Flanker tasks was reduced, and that the peak latencies of P300 and RT were shortened after high-intensity interval training on a treadmill. Since the modulation of P300 and RT were very similar to our findings, their data might be related to the effects of an increasing body temperature as well as the exercise of running on a treadmill. We hypothesized that the effects of an increasing body temperature on P300 differed between the amplitude and latency, paradoxically. Indeed, a study using multi-channel EEG recording showed that the resting-state network topology was significantly correlated with the amplitude of P300, but not the latency [32]. This study suggests that the amplitude and latency of P300 is related to different neural mechanisms, networks, and/or functional connectivity in the whole brain. In future studies, these parameters should be analyzed during passive heat stress.

In addition, we have to consider that changes in these measures during Recovery are due to the monotonous character of the task (i.e., boring effects), because we did not set a time control condition with no passive heat stress. However, in our previous study, we set a time control condition, and subjects performed the auditory oddball paradigms at approximately the same time interval as they did in the Heat stress [9]. We already showed that no significant changes in the behavioral data (RT and error rates) or the peak amplitude and latency of P300 occurred during the time control condition. Thus, a possibility of boring effects can be excluded in the present study.

Judging from the detailed statistics of the peak amplitude and latency of P300, whole-body cooling significantly recovered the amplitude and latency of P300 in Incongruent stimuli, but not fully in Congruent stimuli (Fig 4). These differences may be related to the neural networks and/or regional differences in brain activity between the two stimuli.

## Limitations of the present study

Our ERP methods involving recordings from five electrodes did not directly clarify the brain regions associated with the reduced amplitude of P300 or the accelerated latency of P300.

Therefore, multi-channel ERP recording would be needed to investigate the generator mechanisms, neural networks, and functional connectivity of P300. Furthermore, dehydration associated with passive heat stress may contribute to impaired cognitive function [33]. Although we did not measure levels of dehydration, increased plasma osmolality may contribute to the present observations. Finally, the present study recorded the data from thirteen males and three females, but did not investigate gender differences in the effects of passive heat stress and recovery on the human cognitive function. It is well-known that there are gender differences in thermoregulatory responses [34, 35]. In this study, we controlled the subjects' internal temperature with a water-perfused suit, which minimized individuals' thermoregulatory capacity. Therefore, any gender difference in thermoregulatory responses might affect the cognitive function during the process of recovery from hyperthermia. Further studies are needed to clarify this.

## Conclusion

The present study showed the effects of passive heat stress and recovery on the human cognitive function, measured with ERPs, during visual Flanker tasks. The index of cognitive function, the peak amplitudes of P300 were reduced. An indicator of stimulus classification/evaluation time (peak latency of P300) and the RT were shortened during Heat stress, but such shortening was not noted after whole-body cooling. Our results suggest that hyperthermia affects human cognitive function, reflected by the peak amplitude and latency of P300, but sufficient whole-body cooling recovered such function. In future studies, other preventative strategies, such as drinking and effective cooling of body parts, must be established.

## Supporting information

**S1 Data. Mean values for peak amplitude and latency of P300 during Flanker tasks at all electrodes are shown in S1 Table.**
(XLSX)

**S1 Table. Mean values for peak amplitude and latency of P300 during Flanker tasks with SD.**
(DOCX)

## Acknowledgments

The authors appreciate the time and effort of the volunteer subjects. We thank Mses. Oshiro, Abe, Nagasawa, Matsushita, Urimoto, and Takahara for the recruitment of subjects and their support of this project.

## Author Contributions

**Data curation:** Hiroki Nakata, Manabu Shibasaki.

**Formal analysis:** Hiroki Nakata, Manabu Shibasaki.

**Investigation:** Hiroki Nakata, Manabu Shibasaki.

**Methodology:** Hiroki Nakata, Manabu Shibasaki.

**Project administration:** Hiroki Nakata, Manabu Shibasaki.

**Resources:** Hiroki Nakata, Manabu Shibasaki.

**Supervision:** Ryusuke Kakigi.

**Validation:** Hiroki Nakata, Ryusuke Kakigi, Manabu Shibasaki.

**Writing – original draft:** Hiroki Nakata, Ryusuke Kakigi, Manabu Shibasaki.

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
