## [Decision Letter · Decision Letter 0]

15 Dec 2020

PONE-D-20-33346

Effects of passive heat stress and recovery on human cognitive function: An ERP study

PLOS ONE

Dear Dr. Shibasaki,

Thank you for submitting your manuscript to PLOS ONE. After careful consideration, we feel that it has merit but does not fully meet PLOS ONE’s publication criteria as it currently stands. Therefore, we invite you to submit a revised version of the manuscript that addresses the points raised during the review process.

We look forward to receiving your revised manuscript.

Kind regards,

Daqing Guo

Academic Editor

PLOS ONE

Additional Editor Comments:

This manuscript needs to be significantly improved before it can be accepted for publication in PLoS ONE.

Journal Requirements:

Reviewers' comments:

Reviewer's Responses to Questions

**Comments to the Author**

1. Is the manuscript technically sound, and do the data support the conclusions?

Reviewer #1: Partly

Reviewer #2: Yes

2. Has the statistical analysis been performed appropriately and rigorously? 

Reviewer #1: Yes

Reviewer #2: Yes

3. Have the authors made all data underlying the findings in their manuscript fully available?

Reviewer #1: Yes

Reviewer #2: Yes

4. Is the manuscript presented in an intelligible fashion and written in standard English?

Reviewer #1: Yes

Reviewer #2: Yes

5. Review Comments to the Author

Reviewer #1: In this study the authors explore the effects of passive heat stress and recovery on the human cognitive function with Flanker tasks, involving congruent and incongruent stimuli. The results showed the reaction time (RT) was shortened during Heat rather than Pre and Recovery, and the peak amplitudes of the P300 component in ERPs were significantly smaller during Heat than during Pre, suggesting that hyperthermia alters the human cognitive function. However, I would have below comments for authors to improve the work.

1. In the abstract, there was unclear description on cognitive function, such as “… indicating a recuperated cognitive function…”, “These results suggest that hyperthermia alters the human cognitive function…”. Human cognitive function involves multiple factors, such as attention, memory, decision and so on.

2. In the introduction, “Several non-invasive recording methods have been used to investigate the human cognitive function.”, what’s the non-invasive recording methods? Please give more description.

3. In the last paragraph of the introduction, it looks like the section of discussion, please adjust the paragraph and provide more background introduction.

4. “Sixteen subjects (thirteen males and three females; mean age 21.4 ± 1.0 years) participated in this study”，it reflected the mismatch in the factor of gender. Whether the gender difference has an impact on the effects of passive heat stress and recovery on human cognitive function? Please give more explain.

5. How long does the Flanker task last, and how many trials in total does the task consist of for each subject?

6. In the section of EEG recordings, there only were 5 electrodes, and need make sure the quality of EEG data, what is the impedance of the five electrodes across the whole EEG experiment? And provide the detailed information.

7. In the section Discussion, the logic needs to be strengthened, and provide more supporting materials to support the finding “The effectiveness of cooling on impaired cognitive function in hyper-thermic individuals”.

Reviewer #2: Concentrating on the effect of hyperthermia on human cognition, the authors conducted the Flanker tasks to investigate the differences of ERP and behaviors across different conditions. They found the varied measurements across multiple factors, although these results seem sound, concerns should be resolved by authors before considering for publication.

1. Although the authors mentioned the aim of their study, more information should be given for their novelty and purpose for hyperthermia and related ERP study.

2. Have the authors considered the gender effect on their results, since only three females were included in their study.

3. Since the authors said the ’incompatible’ setting was used in Flanker task, for congruent and incongruent conditions, how the subjects responded to these varied stimuli.

4. Only five channels were recorded, if the authors tried any referencing techniques during their data preprocessing.

5. For statistics, how did the authors perform the multiple comparison correction for ERP and behavior measurements?

6. If it is necessary to present the results for SD of RT when introducing their results.

7. How could the authors explain the confusing results of shorter RT and latency but decreased P300 amplitude for HEAT condition.

8. If it is suitable to using the Flanker task to index the task difficulty, the authors should give some explanation.

9. As depicted, more electrodes would be needed to clarify the brain mechanism accounting for effect of hyperthermia on P300, since our brain works as a large-scale complex network, and concentrating on P300 network, many related works have been implemented, such as

F. Li, et al., Scientific Reports, 2015, 5: 15129; W. Peng, et al., PLoS One, 2012, 7(4):e34163

6. PLOS authors have the option to publish the peer review history of their article (what does this mean?). If published, this will include your full peer review and any attached files.

Reviewer #1: No

Reviewer #2: No

---

## [Author Response · Author response to Decision Letter 0]

20 Jan 2021

Response to Editorial Office

This manuscript needs to be significantly improved before it can be accepted for publication in PLoS ONE.

Journal Requirements:

We revised our manuscript meeting PLOS ONE's style requirements.

We added the ORCiD ID for the corresponding author (Manabu Shibasaki) in the title page.

Response to Reviewer #1

Reviewer #1: In this study the authors explore the effects of passive heat stress and recovery on the human cognitive function with Flanker tasks, involving congruent and incongruent stimuli. The results showed the reaction time (RT) was shortened during Heat rather than Pre and Recovery, and the peak amplitudes of the P300 component in ERPs were significantly smaller during Heat than during Pre, suggesting that hyperthermia alters the human cognitive function. However, I would have below comments for authors to improve the work.

1. In the abstract, there was unclear description on cognitive function, such as “… indicating a recuperated cognitive function…”, “These results suggest that hyperthermia alters the human cognitive function…”. Human cognitive function involves multiple factors, such as attention, memory, decision and so on.

We revised these parts (from page 2, line 47 to page 3, line 58):

“However, the peak amplitudes of the P300 component in ERPs, which involved selective attention, expectancy, and memory updating, were significantly smaller during Heat than during Pre, suggesting the impairment of neural activity in cognitive function. Notably, the peak amplitudes of the P300 component were higher during Recovery than during Heat, indicating that the impaired neural activity had recovered after sufficient whole-body cooling. An indicator of the stimulus classification/evaluation time (peak latency of P300) and the RT were shortened during Heat stress, but such shortening was not noted after whole-body cooling. These results suggest that hyperthermia affects the human cognitive function, reflected by the peak amplitude and latency of the P300 component in ERPs during the Flanker tasks,”

2. In the introduction, “Several non-invasive recording methods have been used to investigate the human cognitive function.”, what’s the non-invasive recording methods? Please give more description.

We revised this part (page 4, lines 93-96):

“Several non-invasive recording methods have been used to investigate the human cognitive function, such as functional magnetic resonance imaging (fMRI), functional near-infrared spectroscopy (fNIRS), and transcranial magnetic stimulation (TMS).”

3. In the last paragraph of the introduction, it looks like the section of discussion, please adjust the paragraph and provide more background introduction.

We revised this part, as suggested (page 6, lines 129-145):

“Thus, we chose “visual” Flanker tasks, which were used to examine the neural systems that resolve the conflict among response options [12]. In these tasks, a central target stimulus is presented simultaneously with distractor stimuli (flankers) and subjects are requested to respond according to the target and ignore the flankers. In general, the reaction time (RT) is shorter with Congruent stimuli than Incongruent stimuli [13, 14], indicating different levels of task difficulty [13, 14]. The second involves recovery after heat stress. As mentioned above, a short period of whole-body cooling after heat stress and face/head cooling during heat stress did not lead to full recovery of the peak amplitude of P300 [9, 10]. In addition, after a thorough literature research, we failed to identify any study examining the recovery process after heat stress by P300. Providing scientific evidence to support recovery from hyperthermia might be important to develop a methodology in our daily life and sports activities. Thus, we should examine whether sufficient whole-body cooling recovers the cognitive function reflected by the P300 component in ERPs.”

4. “Sixteen subjects (thirteen males and three females; mean age 21.4 ± 1.0 years) participated in this study”, it reflected the mismatch in the factor of gender. Whether the gender difference has an impact on the effects of passive heat stress and recovery on human cognitive function? Please give more explain.

Thank you for this suggestion. We added the explanation in limitations of the present study (page 20, lines 466-474):

“Finally, the present study recorded the data from thirteen males and three females, but did not investigate gender differences in the effects of passive heat stress and recovery on the human cognitive function. It is well-known that there are gender differences in thermoregulatory responses [31, 32]. In this study, we controlled the subjects’ internal temperature with a water-perfused suit, which minimized individuals’ thermoregulatory capacity. Therefore, any gender difference in thermoregulatory responses might affect the cognitive function during the process of recovery from hyperthermia. Further studies are needed to clarify this.”

5. How long does the Flanker task last, and how many trials in total does the task consist of for each subject?

We added the explanation (from page 8, line 196 to page 9, line 203):

“A run comprised 160 stimuli (i.e., 8 min), which included 40 stimuli for the left congruent arrowhead, 40 stimuli for the right congruent arrowhead, 40 stimuli for the left incongruent arrowhead, and 40 stimuli for the right incongruent arrowhead. The probability of all stimuli was equal, being presented in a random order. Inter-stimulus interval was fixed. Pre, Heat, and Recovery sessions included 160 stimuli, respectively. In a practice run, subjects were instructed to perform the Flanker tasks with 40 stimuli before recording the Pre session.”

6. In the section of EEG recordings, there only were 5 electrodes, and need make sure the quality of EEG data, what is the impedance of the five electrodes across the whole EEG experiment? And provide the detailed information.

We added the explanation (page 10, lines 236-239):

“All electrodes were detached after ERP recordings in Pre and Heat sessions, respectively, to avoid the effect of sweat on the EEG paste. Just before ERP recordings in Heat and Recovery sessions, all electrodes were again attached.”

7. In the section Discussion, the logic needs to be strengthened, and provide more supporting materials to support the finding “The effectiveness of cooling on impaired cognitive function in hyper-thermic individuals”.

Thank you for this constructive comment. We revised this section (from page 17, line 391 to page 19, line 440).

Response to Reviewer #2

Concentrating on the effect of hyperthermia on human cognition, the authors conducted the Flanker tasks to investigate the differences of ERP and behaviors across different conditions. They found the varied measurements across multiple factors, although these results seem sound, concerns should be resolved by authors before considering for publication.

1. Although the authors mentioned the aim of their study, more information should be given for their novelty and purpose for hyperthermia and related ERP study.

Thank you for this suggestion. We revised the Introduction section (page 6, lines 129-145):

“Thus, we chose “visual” Flanker tasks, which were used to examine the neural systems that resolve the conflict among response options [12]. In these tasks, a central target stimulus is presented simultaneously with distractor stimuli (flankers) and subjects are requested to respond according to the target and ignore the flankers. In general, the reaction time (RT) is shorter with Congruent stimuli than Incongruent stimuli [13, 14], indicating different levels of task difficulty [13, 14]. The second involves recovery after heat stress. As mentioned above, a short period of whole-body cooling after heat stress and face/head cooling during heat stress did not lead to full recovery of the peak amplitude of P300 [9, 10]. In addition, after a thorough literature research, we failed to identify any study examining the recovery process after heat stress by P300. Providing scientific evidence to support recovery from hyperthermia might be important to develop a methodology in our daily life and sports activities. Thus, we should examine whether sufficient whole-body cooling recovers the cognitive function reflected by the P300 component in ERPs.”

2. Have the authors considered the gender effect on their results, since only three females were included in their study.

Thank you for this suggestion. Reviewer #1 also pointed out the same problem. We added the explanation in limitations of the present study (page 20, lines 466-474):

“Finally, the present study recorded the data from thirteen males and three females, but did not investigate gender differences in the effects of passive heat stress and recovery on the human cognitive function. It is well-known that there are gender differences in thermoregulatory responses [31, 32]. In this study, we controlled the subjects’ internal temperature with a water-perfused suit, which minimized individuals’ thermoregulatory capacity. Therefore, any gender difference in thermoregulatory responses might affect the cognitive function during the process of recovery from hyperthermia. Further studies are needed to clarify this.”

3. Since the authors said the ’incompatible’ setting was used in Flanker task, for congruent and incongruent conditions, how the subjects responded to these varied stimuli.

We added the explanation (page 8, lines 188-194):

“For example, when the central arrowhead was directed to the left, the subjects were instructed to press a button with their left thumb. However, after being trained in the compatible setting, subjects were requested to respond in a direction opposite to the target arrowhead, called an ‘incompatible’ setting. For example, when the central arrowhead was directed to the left, the subjects were instructed to press a button with their right thumb.”

4. Only five channels were recorded, if the authors tried any referencing techniques during their data preprocessing.

Reviewer #1 also pointed out the same problem. We added the explanation (page 10, lines 230-231):

“Each scalp electrode was referenced to linked earlobes, which were mathematically calculated and averaged for reference.”

5. For statistics, how did the authors perform the multiple comparison correction for ERP and behavior measurements?

We performed the Bonferroni post-hoc multiple-comparison to identify the specific differences among sessions, but did not adjust the comparison correction. We added the explanation (from page 11, line 267 to page 12, line 277):

“For all repeated-measures factors, we tested whether Mauchly’s sphericity assumption was violated. If the result of Mauchly’s test was significant and the assumption of sphericity was violated, Greenhouse-Geisser adjustment was used to correct sphericity by altering the degrees of freedom using a correction coefficient epsilon. When a significant effect of Session was identified, the Bonferroni post-hoc multiple-comparison was adjusted to identify the specific differences among sessions. When a significant effect of Stimulus was identified, the post-hoc paired-t-test was adjusted to identify the specific difference between Congruent and Incongruent stimuli. We did not adjust the multiple comparison correction for all data.”

6. If it is necessary to present the results for SD of RT when introducing their results.

Thank you for this helpful comment. We added the explanation on the results for SD of RT in many parts of Discussion sections.

7. How could the authors explain the confusing results of shorter RT and latency but decreased P300 amplitude for HEAT condition.

We revised the Discussion section (from page 18, line 424 to page 19, line 440):

“We considered that acceleration of RT and the latency of P300 and small response variability were related to specific effective connectivity in neural networks with an increasing internal temperature, which differs from the theory of structural connectivity [see a review, 27]. Kao et al. [28] recently reported that the peak amplitude of P300 in Flanker tasks was reduced, and that the peak latencies of P300 and RT were shortened after high-intensity interval training on a treadmill. Since the modulation of P300 was very similar to our findings, their data might be related to the effects of an increasing body temperature as well as the exercise of running on a treadmill. We hypothesized that the effects of an increasing body temperature on P300 differed between the amplitude and latency, paradoxically. Indeed, a study using multi-channel EEG recording showed that the resting-state network topology was significantly correlated with the amplitude of P300, but not the latency [29]. This study suggests that the amplitude and latency of P300 is related to different neural mechanisms, networks, and/or functional connectivity in the whole brain.”

8. If it is suitable to using the Flanker task to index the task difficulty, the authors should give some explanation.

We added the Introduction section to explain the task difficulty in the Flanker task (page 6, lines 129-135):

“Thus, we chose “visual” Flanker tasks, which were used to examine the neural systems that resolve the conflict among response options [12]. In these tasks, a central target stimulus is presented simultaneously with distractor stimuli (flankers) and subjects are requested to respond according to the target and ignore the flankers. In general, the reaction time (RT) is shorter with Congruent stimuli than Incongruent stimuli [13, 14], indicating different levels of task difficulty [13, 14].”

9. As depicted, more electrodes would be needed to clarify the brain mechanism accounting for effect of hyperthermia on P300, since our brain works as a large-scale complex network, and concentrating on P300 network, many related works have been implemented, such as

> F. Li, et al., Scientific Reports, 2015, 5: 15129; W. Peng, et al., PLoS One, 2012, 7(4):e34163

Thank you for this suggestion. We added the explanation in Discussion (from page 18, line 433 to page 19, line 440):

“We hypothesized that the effects of an increasing body temperature on P300 differed between the amplitude and latency, paradoxically. Indeed, a study using multi-channel EEG recording showed that the resting-state network topology was significantly correlated with the amplitude of P300, but not the latency [29]. This study suggests that the amplitude and latency of P300 is related to different neural mechanisms, networks, and/or functional connectivity in the whole brain.”

Limitations of the present study (from page 19, line 458 to page 20, line 462):

“Our ERP methods involving recordings from five electrodes did not directly clarify the brain regions associated with the reduced amplitude of P300 or the accelerated latency of P300. Therefore, multi-channel ERP recording would be needed to investigate the generator mechanisms, neural networks, and functional connectivity of P300.”

---

## [Decision Letter · Decision Letter 1]

15 Mar 2021

PONE-D-20-33346R1

Effects of passive heat stress and recovery on human cognitive function: An ERP study

PLOS ONE

Dear Dr. Shibasaki,

Thank you for submitting your manuscript to PLOS ONE. After careful consideration, we feel that it has merit but does not fully meet PLOS ONE’s publication criteria as it currently stands. Therefore, we invite you to submit a revised version of the manuscript that addresses the points raised during the review process.

First, allow me to introduce myself as the new editor handling your paper, as requested by the editorial office of PlosONE. Since I was not involved in the review process of your manuscript from the start, I have requested an evaluation from a third expert, to make sure that issues that might have been missed in the first round of reviews have been rechecked. I am pleased to say that the feedback on your paper is overall very positive, as is my own evaluation. The study is very interesting and the data and analyses provided are overall convincing.

As you will see, the two reviewers who originally reviewed your paper have now recommended acceptance and require no other changes to the paper. Reviewer 3, however, has some concerns that would need to be addressed before I can recommend your paper for publication in PlosONE. At the moment, and unless the revisions requested bring up a critical issue in the next round, I expect to be in a position to make a final decision on your submission without requiring another round of external reviews. Whilst points 1 and 2 raised by the reviewer 3 are mainly interpretative and methodological, which should be quite straightforward to address, point 3 is more delicate. I invite you to address the remaining issues raised by Reviewer 3 carefully and submit your revised paper at your earliest convenience.

We look forward to receiving your revised manuscript.

Kind regards,

Guillaume Thierry, Ph.D.

Academic Editor

PLOS ONE

Journal Requirements:

Reviewers' comments:

Reviewer's Responses to Questions

**Comments to the Author**

1. If the authors have adequately addressed your comments raised in a previous round of review and you feel that this manuscript is now acceptable for publication, you may indicate that here to bypass the “Comments to the Author” section, enter your conflict of interest statement in the “Confidential to Editor” section, and submit your "Accept" recommendation.

Reviewer #1: All comments have been addressed

Reviewer #2: All comments have been addressed

Reviewer #3: (No Response)

2. Is the manuscript technically sound, and do the data support the conclusions?

Reviewer #1: (No Response)

Reviewer #2: Yes

Reviewer #3: Partly

3. Has the statistical analysis been performed appropriately and rigorously? 

Reviewer #1: (No Response)

Reviewer #2: Yes

Reviewer #3: Yes

4. Have the authors made all data underlying the findings in their manuscript fully available?

Reviewer #1: (No Response)

Reviewer #2: Yes

Reviewer #3: Yes

5. Is the manuscript presented in an intelligible fashion and written in standard English?

Reviewer #1: (No Response)

Reviewer #2: Yes

Reviewer #3: Yes

6. Review Comments to the Author

Reviewer #1: The authors have made substantial changes to the manuscript to respond to my previously comments. I have no other comments.

Reviewer #2: All of my concerns have been resolved by authors, and I do not have any further comments. Thanks for the authors' work.

Reviewer #3: The manuscript reports analyses of ERP data before, during, and after participants underwent a heat stress challenge paradigm. These data must have been demanding to acquire and the authors are to be congratulated on this. I have a few points I would like them to respond to, a few substantive issues followed by more minor points:

1. The changes to cognitive function during heat stress and recovery are framed as impairments in the discussion, but the actual task performance, in terms of RT and consistency, is improved, while accuracy appears similar across conditions (perhaps a ceiling effect?). Yes the P300 was smaller in amplitude, but I don’t think that is evidence of impairment in of itself. This needs some attention in the Discussion.

2. I don’t see any mention of artefact correction or rejection – were all trials used? If EEG data quality differed between conditions, that could plausibly drive effects in component amplitudes – are you able to assess how likely that is to be a problem?

3. Related to the above point, you mention that electrodes were detached between conditions to minimise sweat artefacts. Can you reassure me that this dealt with this, as this does seem like a potentially important issue?

4. The theoretical relevance of changes to variability of RTs is not discussed.

Minor issues

1. Breaking the introduction into more paragraphs might make it a bit more reader friendly.

2. Page 7 – ‘The man body mass and height…’ – Should ‘man’ read ‘mean’?

3. Page 8 – ‘the 2nd ERP was recorded after…’ – I would change to ‘ERP session was’ or ‘ERPs were’. Likewise for ‘the 3rd ERP

4. Page 12 – ‘Significance was set at p < 0.05.’ – ‘significance’ should read ‘alpha’

7. PLOS authors have the option to publish the peer review history of their article (what does this mean?). If published, this will include your full peer review and any attached files.

Reviewer #1: No

Reviewer #2: No

Reviewer #3: **Yes: **Christopher W N Saville

---

## [Author Response · Author response to Decision Letter 1]

23 Mar 2021

Response to Reviewer #3

The manuscript reports analyses of ERP data before, during, and after participants underwent a heat stress challenge paradigm. These data must have been demanding to acquire and the authors are to be congratulated on this. I have a few points I would like them to respond to, a few substantive issues followed by more minor points:

1. The changes to cognitive function during heat stress and recovery are framed as impairments in the discussion, but the actual task performance, in terms of RT and consistency, is improved, while accuracy appears similar across conditions (perhaps a ceiling effect?). Yes the P300 was smaller in amplitude, but I don’t think that is evidence of impairment in of itself. This needs some attention in the Discussion.

We added the explanation in the Discussion (page 19, lines 437-455):

“The interpretation of these results requires special attention. We considered that acceleration of RT and the latency of P300 and smaller SD of RT (response variability) were related to specific effective connectivity in neural networks with an increasing internal temperature, which differs from the theory of structural connectivity [see a review, 30]. Kao et al. [31] recently reported that the peak amplitude of P300 in Flanker tasks was reduced, and that the peak latencies of P300 and RT were shortened after high-intensity interval training on a treadmill. Since the modulation of P300 and RT were very similar to our findings, their data might be related to the effects of an increasing body temperature as well as the exercise of running on a treadmill. We hypothesized that the effects of an increasing body temperature on P300 differed between the amplitude and latency, paradoxically. Indeed, a study using multi-channel EEG recording showed that the resting-state network topology was significantly correlated with the amplitude of P300, but not the latency [32]. This study suggests that the amplitude and latency of P300 is related to different neural mechanisms, networks, and/or functional connectivity in the whole brain. In future studies, these parameters should be analyzed during passive heat stress.”

2. I don’t see any mention of artefact correction or rejection – were all trials used? If EEG data quality differed between conditions, that could plausibly drive effects in component amplitudes – are you able to assess how likely that is to be a problem?

We added the explanation in Materials and Methods section (page 10, lines 235-237):

“Artifacts or noise caused by blinking or sweating were excluded on-line. If the number averaged was less than 35 for each stimulus, additional trials were performed.”

3. Related to the above point, you mention that electrodes were detached between conditions to minimise sweat artefacts. Can you reassure me that this dealt with this, as this does seem like a potentially important issue?

Thank you for constructive comment. We added the discussion on this issue (page 17, lines 390-399):

“In addition, the reduced amplitude of P300 might be related to electrical noise due to sweating during passive heat stress. However, as mentioned above, to avoid the effect of sweat on the EEG paste, we reattached all electrodes just before ERP recordings in Heat and Recovery sessions. Our previous study using the same procedure as in the present study showed that the amplitudes of some components in somatosensory evoked potentials (SEPs) reduced during passive heat stress, but other components did not [22, 23]. If the presence of sweat influences the SEP waveforms, all SEP components should be distorted. Therefore, we considered that sweat had a reduced effect on the amplitude of P300 in the present study.”

4. The theoretical relevance of changes to variability of RTs is not discussed.

We added the discussion (from page 18, line 430 to page 19, line 455).

Minor issues

1. Breaking the introduction into more paragraphs might make it a bit more reader friendly.

We revised the introduction section.

2. Page 7 – ‘The man body mass and height…’ – Should ‘man’ read ‘mean’?

Revised (page 7, line 155):

“The mean body mass and height of the subjects were 71.7 ± 11.3 kg, and 170.5 ± 6.3 cm, respectively.”

3. Page 8 – ‘the 2nd ERP was recorded after…’ – I would change to ‘ERP session was’ or ‘ERPs were’. Likewise for ‘the 3rd ERP

Revised (page 8, line 175):

“ERPs were recorded after the external ear canal temperature”

Revised (page 8, line 180):

“ERPs were recorded (Recovery session) (Fig 1A).”

4. Page 12 – ‘Significance was set at p < 0.05.’ – ‘significance’ should read ‘alpha’

Revised (page 12, line 281):

“All statistical analyses were conducted with α = 0.05.”

---

## [Decision Letter · Decision Letter 2]

5 Jul 2021

Effects of passive heat stress and recovery on human cognitive function: An ERP study

PONE-D-20-33346R2

Dear Dr. Shibasaki,

We’re pleased to inform you that your manuscript has been judged scientifically suitable for publication and will be formally accepted for publication once it meets all outstanding technical requirements.

Kind regards,

Thalia Fernandez, Ph.D.

Academic Editor

PLOS ONE

Additional Editor Comments (optional):

Reviewers' comments:

Reviewer's Responses to Questions

**Comments to the Author**

1. If the authors have adequately addressed your comments raised in a previous round of review and you feel that this manuscript is now acceptable for publication, you may indicate that here to bypass the “Comments to the Author” section, enter your conflict of interest statement in the “Confidential to Editor” section, and submit your "Accept" recommendation.

Reviewer #1: All comments have been addressed

Reviewer #2: All comments have been addressed

Reviewer #3: (No Response)

2. Is the manuscript technically sound, and do the data support the conclusions?

Reviewer #1: Yes

Reviewer #2: Yes

Reviewer #3: Yes

3. Has the statistical analysis been performed appropriately and rigorously? 

Reviewer #1: Yes

Reviewer #2: N/A

Reviewer #3: Yes

4. Have the authors made all data underlying the findings in their manuscript fully available?

Reviewer #1: Yes

Reviewer #2: No

Reviewer #3: Yes

5. Is the manuscript presented in an intelligible fashion and written in standard English?

Reviewer #1: Yes

Reviewer #2: Yes

Reviewer #3: Yes

6. Review Comments to the Author

Reviewer #1: The authors have made substantial changes to the manuscript to respond to my previously comments. I have no further comments.

Reviewer #2: Thanks for the authors hard-working, no further comments for this work, so I suggest the publication of this work.

Reviewer #3: Thank you for addressing my comments. However, the conclusions section still claims that cognition is impaired. Both by use of the word 'impairment' and by talking about recovery after cooling. I am not convinced the data support this characterisation so ask suggest the authors to either amend the conclusions or make the case why they think their data support a claim of impairment. Otherwise I am happy with the manuscript.

7. PLOS authors have the option to publish the peer review history of their article (what does this mean?). If published, this will include your full peer review and any attached files.

Reviewer #1: No

Reviewer #2: No

Reviewer #3: **Yes: **Christopher W N Saville

---

## [Editor Report · Acceptance letter]

12 Jul 2021

PONE-D-20-33346R2 

Effects of passive heat stress and recoveryon human cognitive function: An ERP study 

Dear Dr. Shibasaki:

I'm pleased to inform you that your manuscript has been deemed suitable for publication in PLOS ONE. Congratulations! Your manuscript is now with our production department. 

Kind regards, 

on behalf of

Dr. Thalia Fernandez 

Academic Editor

PLOS ONE